# Rigorous Dynamics and Consistent Estimation in Arbitrarily Conditioned Linear Systems

**Alyson K. Fletcher**
Dept. Statistics
UC Los Angeles
akfletcher@ucla.edu

**Mojtaba Sahraee-Ardakan**
Dept. EE,
UC Los Angeles
msahraee@ucla.edu

**Sundeep Rangan**
Dept. ECE,
NYU
srangan@nyu.edu

**Philip Schniter**
Dept. ECE,
The Ohio State Univ.
schniter@ece.osu.edu

## Abstract

We consider the problem of estimating a random vector $\mathbf{x}$ from noisy linear measurements $\mathbf{y} = \mathbf{A}\mathbf{x} + \mathbf{w}$ in the setting where parameters $\boldsymbol{\theta}$ on the distribution of $\mathbf{x}$ and $\mathbf{w}$ must be learned in addition to the vector $\mathbf{x}$. This problem arises in a wide range of statistical learning and linear inverse problems. Our main contribution shows that a computationally simple iterative message passing algorithm can provably obtain asymptotically consistent estimates in a certain high-dimensional large system limit (LSL) under very general parametrizations. Importantly, this LSL applies to all right-rotationally random $\mathbf{A}$ – a much larger class of matrices than i.i.d. sub-Gaussian matrices to which many past message passing approaches are restricted. In addition, a simple testable condition is provided in which the mean square error (MSE) on the vector $\mathbf{x}$ matches the Bayes optimal MSE predicted by the replica method. The proposed algorithm uses a combination of Expectation-Maximization (EM) with a recently-developed Vector Approximate Message Passing (VAMP) technique. We develop an analysis framework that shows that the parameter estimates in each iteration of the algorithm converge to deterministic limits that can be precisely predicted by a simple set of state evolution (SE) equations. The SE equations, which extends those of VAMP without parameter adaptation, depend only on the initial parameter estimates and the statistical properties of the problem and can be used to predict consistency and precisely characterize other performance measures of the method.

## 1 Introduction

Consider the problem of estimating a random vector $\mathbf{x}^0$ from linear measurements $\mathbf{y}$ of the form

$$\mathbf{y} = \mathbf{A}\mathbf{x}^0 + \mathbf{w}, \quad \mathbf{w} \sim \mathcal{N}(\mathbf{0}, \theta_2^{-1}\mathbf{I}), \quad \mathbf{x}^0 \sim p(\mathbf{x}|\boldsymbol{\theta}_1), \tag{1}$$

where $\mathbf{A} \in \mathbb{R}^{M \times N}$ is a known matrix, $p(\mathbf{x}|\boldsymbol{\theta}_1)$ is a density on $\mathbf{x}^0$ with parameters $\boldsymbol{\theta}_1$, $\mathbf{w}$ is additive white Gaussian noise (AWGN) independent of $\mathbf{x}^0$, and $\theta_2 > 0$ is the noise precision (inverse variance). The goal is to estimate $\mathbf{x}^0$ along with simultaneously learning the unknown parameters $\boldsymbol{\theta} := (\boldsymbol{\theta}_1, \theta_2)$ from the data $\mathbf{y}$ and $\mathbf{A}$. This problem arises in Bayesian forms of linear inverse problems in signal processing, as well as in linear regression in statistics.

Exact estimation of the parameters $\boldsymbol{\theta}$ via maximum likelihood or other methods is generally intractable. One promising class of approximate methods combines approximate message passing (AMP) [1]

with expectation-maximization (EM). AMP and its generalizations [2] are a powerful, relatively recent, class of algorithms based on expectation propagation-type techniques. The AMP methodology has the benefit of being computationally fast and has been successfully applied to a wide range of problems. Most importantly, for large, i.i.d., sub-Gaussian random matrices $\mathbf{A}$, the performance of AMP methods can be exactly predicted by a scalar *state evolution* (SE) [3, 4] that provides testable conditions for optimality, even for non-convex priors. When the parameters $\boldsymbol{\theta}$ are unknown, AMP can be easily combined with EM for joint learning of the parameters $\boldsymbol{\theta}$ and vector $\mathbf{x}$ [5–7].

A recent work [8] has combined EM with the so-called Vector AMP (VAMP) method of [9]. Similar to AMP, VAMP is based on expectation propagation (EP) approximations of belief propagation [10, 11] and can also be considered as a special case of expectation consistent (EC) approximate inference [12–14]. VAMP's key attraction is that it applies to a larger class of matrices $\mathbf{A}$ than standard AMP methods. Aside from Gaussian i.i.d. $\mathbf{A}$, standard AMP techniques often diverge and require a variety of modifications for stability [15–18]. In contrast, VAMP has provable SE analyses and convergence guarantees that apply to all right-rotationally invariant matrices $\mathbf{A}$ [9, 19] – a significantly larger class of matrices than i.i.d. Gaussians. Under further conditions, the mean-squared error (MSE) of VAMP matches the replica predictions for optimality [20–23]. For the case when the distribution on $\mathbf{x}$ and $\mathbf{w}$ are unknown, the work [8] proposed to combine EM and VAMP using the approximate inference framework of [24]. The combination of AMP with EM methods have been particularly successful in neural modeling problems [25, 26]. While [8] provides numerical simulations demonstrating excellent performance of this EM-VAMP method on a range of synthetic data, there were no provable convergence guarantees.

**Contributions of this work**    The SE analysis thus provides a rigorous and exact characterization of the dynamics of EM-VAMP. In particular, the analysis can determine under which initial conditions and problem statistics EM-VAMP will yield asymptotically consistent parameter estimates.

- *Rigorous state evolution analysis:* We provide a rigorous analysis of a generalization of EM-VAMP that we call Adaptive VAMP. Similar to the analysis of VAMP, we consider a certain large system limit (LSL) where the matrix $\mathbf{A}$ is random and right-rotationally invariant. Importantly, this class of matrices is much more general than i.i.d. Gaussians used in the original LSL analysis of Bayati and Montanari [3]. It is shown (Theorem 1) that in the LSL, the parameter estimates at each iteration converge to deterministic limits $\overline{\theta}_k$ that can be computed from a set of SE equations that extend those of VAMP. The analysis also exactly characterizes the asymptotic joint distribution of the estimates $\widehat{\mathbf{x}}$ and the true vector $\mathbf{x}^0$. The SE equations depend only on the initial parameter estimate, the adaptation function, and statistics on the matrix $\mathbf{A}$, the vector $\mathbf{x}^0$ and noise $\mathbf{w}$.

- *Asymptotic consistency*: It is also shown (Theorem 2) that under an additional identifiability condition and a simple auto-tuning procedure, Adaptive VAMP can yield provably consistent parameter estimates in the LSL. The technique uses an ML estimation approach from [7]. Remarkably, the result is true under very general problem formulations.

- *Bayes optimality*: In the case when the parameter estimates converge to the true value, the behavior of adaptive VAMP matches that of VAMP. In this case, it is shown in [9] that, when the SE equations have a unique fixed point, the MSE of VAMP matches the MSE of the Bayes optimal estimator predicted by the replica method [21–23].

In this way, we have developed a computationally efficient method for a large class of linear inverse problems with the properties that, in a certain high-dimensional limit: (1) the performance of the algorithm can be exactly characterized, (2) the parameter estimates $\widehat{\boldsymbol{\theta}}$ are asymptotically consistent; and (3) the algorithm has testable conditions for which the signal estimates $\widehat{\mathbf{x}}$ match replica predictions for Bayes optimality.

## 2   VAMP with Adaptation

Assume the prior on $\mathbf{x}$ can be written as

$$p(\mathbf{x}|\boldsymbol{\theta}_1) = \frac{1}{Z_1(\boldsymbol{\theta}_1)} \exp\left[-f_1(\mathbf{x}|\boldsymbol{\theta}_1)\right], \quad f_1(\mathbf{x}|\boldsymbol{\theta}_1) = \sum_{n=1}^{N} f_1(x_n|\boldsymbol{\theta}_1), \tag{2}$$

---
**Algorithm 1** Adaptive VAMP
---
**Require:** Matrix $\mathbf{A} \in \mathbb{R}^{M \times N}$, measurement vector $\mathbf{y}$, denoiser function $\mathbf{g}_1(\cdot)$, statistic function $\phi_1(\cdot)$, adaptation function $T_1(\cdot)$ and number of iterations $N_{\mathrm{it}}$.

1: Select initial $\mathbf{r}_{10}$, $\gamma_{10} \geq 0$, $\widehat{\boldsymbol{\theta}}_{10}, \widehat{\theta}_{20}$.
2: **for** $k = 0, 1, \ldots, N_{\mathrm{it}} - 1$ **do**
3:     // Input denoising
4:     $\widehat{\mathbf{x}}_{1k} = \mathbf{g}_1(\mathbf{r}_{1k}, \gamma_{1k}, \widehat{\boldsymbol{\theta}}_{1k}))$,       $\eta_{1k}^{-1} = \gamma_{1k}/\langle \mathbf{g}_1'(\mathbf{r}_{1k}, \gamma_{1k}, \widehat{\boldsymbol{\theta}}_{1k}) \rangle$
5:     $\gamma_{2k} = \eta_{1k} - \gamma_{1k}$
6:     $\mathbf{r}_{2k} = (\eta_{1k}\widehat{\mathbf{x}}_{1k} - \gamma_{1k}\mathbf{r}_{1k})/\gamma_{2k}$
7:
8:     // Input parameter update
9:     $\widehat{\boldsymbol{\theta}}_{1,k+1} = T_1(\mu_{1k})$,       $\mu_{1k} = \langle \phi_1(\mathbf{r}_{1k}, \gamma_{1k}, \widehat{\boldsymbol{\theta}}_{1k}) \rangle$
10:
11:     // Output estimation
12:     $\widehat{\mathbf{x}}_{2k} = \mathbf{Q}_k^{-1}(\widehat{\theta}_{2k}\mathbf{A}^{\mathsf{T}}\mathbf{y} + \gamma_{2k}\mathbf{r}_{2k})$,       $\mathbf{Q}_k = \widehat{\theta}_{2k}\mathbf{A}^{\mathsf{T}}\mathbf{A} + \gamma_{2k}\mathbf{I}$
13:     $\eta_{2k}^{-1} = (1/N)\operatorname{tr}(\mathbf{Q}_k^{-1})$
14:     $\gamma_{1,k+1} = \eta_{2k} - \gamma_{2k}$
15:     $\mathbf{r}_{1,k+1} = (\eta_{2k}\widehat{\mathbf{x}}_{2k} - \gamma_{2k}\mathbf{r}_{2k})/\gamma_{1,k+1}$
16:
17:     // Output parameter update
18:     $\widehat{\theta}_{2,k+1}^{-1} = (1/N)\{\|\mathbf{y} - \mathbf{A}\widehat{\mathbf{x}}_{2k}\|^2 + \operatorname{tr}(\mathbf{A}\mathbf{Q}_k^{-1}\mathbf{A}^{\mathsf{T}})\}$
19: **end for**
---

where $f_1(\cdot)$ is a separable penalty function, $\boldsymbol{\theta}_1$ is a parameter vector and $Z_1(\boldsymbol{\theta}_1)$ is a normalization constant. With some abuse of notation, we have used $f_1(\cdot)$ for the function on the vector $\mathbf{x}$ and its components $x_n$. Since $f_1(\mathbf{x}|\boldsymbol{\theta}_1)$ is separable, $\mathbf{x}$ has i.i.d. components conditioned on $\boldsymbol{\theta}_1$. The likelihood function under the Gaussian model (1) can be written as

$$p(\mathbf{y}|\mathbf{x}, \theta_2) := \frac{1}{Z_2(\theta_2)} \exp\left[-f_2(\mathbf{x}, \mathbf{y}|\theta_2)\right], \quad f_2(\mathbf{x}, \mathbf{y}|\theta_2) := \frac{\theta_2}{2}\|\mathbf{y} - \mathbf{A}\mathbf{x}\|^2, \tag{3}$$

where $Z_2(\theta_2) = (2\pi/\theta_2)^{N/2}$. The joint density of $\mathbf{x}, \mathbf{y}$ given parameters $\boldsymbol{\theta} = (\boldsymbol{\theta}_1, \theta_2)$ is then

$$p(\mathbf{x}, \mathbf{y}|\boldsymbol{\theta}) = p(\mathbf{x}|\boldsymbol{\theta}_1)p(\mathbf{y}|\mathbf{x}, \theta_2). \tag{4}$$

The problem is to estimate the parameters $\boldsymbol{\theta} = (\boldsymbol{\theta}_1, \theta_2)$ along with the vector $\mathbf{x}^0$.

The steps of the proposed adaptive VAMP algorithm to perform this estimation are shown in Algorithm 1, which is a generalization of the the EM-VAMP method in [8]. In each iteration, the algorithm produces, for $i = 1, 2$, estimates $\widehat{\theta}_i$ of the parameter $\theta_i$, along with estimates $\widehat{\mathbf{x}}_{ik}$ of the vector $\mathbf{x}^0$. The algorithm is tuned by selecting three key functions: (i) a *denoiser function* $\mathbf{g}_1(\cdot)$; (ii) an *adaptation statistic* $\phi_1(\cdot)$; and (iii) a *parameter selection function* $T_1(\cdot)$. The denoiser is used to produce the estimates $\widehat{\mathbf{x}}_{1k}$, while the adaptation statistic and parameter estimation functions produce the estimates $\widehat{\boldsymbol{\theta}}_{1k}$.

**Denoiser function** The denoiser function $\mathbf{g}_1(\cdot)$ is discussed in detail in [9] and is generally based on the prior $p(\mathbf{x}|\boldsymbol{\theta}_1)$. In the original EM-VAMP algorithm [8], $\mathbf{g}_1(\cdot)$ is selected as the so-called minimum mean-squared error (MMSE) denoiser. Specifically, in each iteration, the variables $\mathbf{r}_i$, $\gamma_i$ and $\widehat{\theta}_i$ were used to construct *belief estimates*,

$$b_i(\mathbf{x}|\mathbf{r}_i, \gamma_i, \widehat{\theta}_i) \propto \exp\left[-f_i(\mathbf{x}, \mathbf{y}|\widehat{\theta}_i) - \frac{\gamma_i}{2}\|\mathbf{x} - \mathbf{r}_i\|^2\right], \tag{5}$$

which represent estimates of the posterior density $p(\mathbf{x}|\mathbf{y}, \boldsymbol{\theta})$. To keep the notation symmetric, we have written $f_1(\mathbf{x}, \mathbf{y}|\widehat{\boldsymbol{\theta}}_1)$ for $f_1(\mathbf{x}|\widehat{\boldsymbol{\theta}}_1)$ even though the first penalty function does not depend on $\mathbf{y}$. The EM-VAMP method then selects $\mathbf{g}_1(\cdot)$ to be the mean of the belief estimate,

$$\mathbf{g}_1(\mathbf{r}_1, \gamma_1, \boldsymbol{\theta}_1) := \mathbb{E}\left[\mathbf{x}|\mathbf{r}_1, \gamma_1, \boldsymbol{\theta}_1\right]. \tag{6}$$

For line 4 of Algorithm 1, we define $[\mathbf{g}_1'(\mathbf{r}_{1k}, \gamma_{1k}, \boldsymbol{\theta}_1)]_n := \partial[\mathbf{g}_1(\mathbf{r}_{1k}, \gamma_{1k}, \boldsymbol{\theta}_1)]_n/\partial r_{1n}$ and we use $\langle \cdot \rangle$ for the empirical mean of a vector, i.e., $\langle \mathbf{u} \rangle = (1/N)\sum_{n=1}^{N} u_n$. Hence, $\eta_{1k}$ in line 4 is a scaled

inverse divergence. It is shown in [9] that, for the MMSE denoiser (6), $\eta_{1k}$ is the inverse average posterior variance.

**Estimation for $\theta_1$ with finite statistics** For the EM-VAMP algorithm [8], the parameter update for $\widehat{\theta}_{1,k+1}$ is performed via a maximization

$$\widehat{\theta}_{1,k+1} = \arg\max_{\theta_1} \mathbb{E}\left[\ln p(\mathbf{x}|\theta_1)\,\Big|\,\mathbf{r}_{1k}, \gamma_{1k}, \widehat{\theta}_{1k}\right], \tag{7}$$

where the expectation is with respect to the belief estimate $b_i(\cdot)$ in (5). It is shown in [8] that using (7) is equivalent to an approximation of the M-step in the standard EM method. In the adaptive VAMP method in Algorithm 1, the M-step maximization (7) is replaced by line 9. Note that line 9 again uses $\langle\cdot\rangle$ to denote empirical average,

$$\mu_{1k} = \langle\phi_1(\mathbf{r}_{1k}, \gamma_{1k}, \widehat{\theta}_{1k})\rangle := \frac{1}{N}\sum_{n=1}^{N}\phi_1(r_{1k,n}, \gamma_{1k}, \widehat{\theta}_{1k}) \in \mathbb{R}^d, \tag{8}$$

so $\mu_{1k}$ is the empirical average of some $d$-dimensional statistic $\phi_1(\cdot)$ over the components of $\mathbf{r}_{1k}$. The parameter estimate update $\widehat{\theta}_{1,k+1}$ is then computed from some function of this statistic, $T_1(\mu_{1k})$.

We show in the full paper [27] that there are two important cases where the EM update (7) can be computed from a finite-dimensional statistic as in line 9: (i) The prior $p(\mathbf{x}|\theta_1)$ is given by an exponential family, $f_1(\mathbf{x}|\theta_1) = \theta_1^\mathsf{T}\varphi(\mathbf{x})$ for some sufficient statistic $\varphi(\mathbf{x})$; and (ii) There are a finite number of values for the parameter $\theta_1$. For other cases, we can approximate more general parametrizations via discretization of the parameter values $\vec{\theta}_1$. The updates in line 9 can also incorporate other types of updates as we will see below. But, we stress that it is preferable to compute the estimate for $\theta_1$ directly from the maximization (7) – the use of a finite-dimensional statistic is for the sake of analysis.

**Estimation for $\theta_2$ with finite statistics** It will be useful to also write the adaptation of $\theta_2$ in line 18 of Algorithm 1 in a similar form as line 9. First, take a singular value decomposition (SVD) of $\mathbf{A}$ of the form

$$\mathbf{A} = \mathbf{U}\mathbf{S}\mathbf{V}^\mathsf{T}, \quad \mathbf{S} = \mathrm{Diag}(\mathbf{s}), \tag{9}$$

and define the transformed error and transformed noise,

$$\mathbf{q}_k := \mathbf{V}^\mathsf{T}(\mathbf{r}_{2k} - \mathbf{x}^0), \quad \boldsymbol{\xi} := \mathbf{U}^\mathsf{T}\mathbf{w}. \tag{10}$$

Then, it is shown in the full paper [27] that $\widehat{\theta}_{2,k+1}$ in line 18 can be written as

$$\widehat{\theta}_{2,k+1} = T_2(\mu_{2k}) := \frac{1}{\mu_{2k}}, \quad \mu_{2k} = \langle\phi_2(\mathbf{q}_2, \boldsymbol{\xi}, \mathbf{s}, \gamma_{2k}, \widehat{\theta}_{2k})\rangle \tag{11}$$

where

$$\phi_2(q, \xi, s, \gamma_2, \widehat{\theta}_2) := \frac{\gamma_2^2}{(s^2\widehat{\theta}_2 + \gamma_2)^2}(sq + \xi)^2 + \frac{s^2}{s^2\widehat{\theta}_2 + \gamma_2}. \tag{12}$$

Of course, we cannot directly compute $\mathbf{q}_k$ in (10) since we do not know the true $\mathbf{x}^0$. Nevertheless, this form will be useful for analysis.

## 3 State Evolution in the Large System Limit

### 3.1 Large System Limit

Similar to the analysis of VAMP in [9], we analyze Algorithm 1 in a certain large system limit (LSL). The LSL framework was developed by Bayati and Montanari in [3] and we review some of the key definitions in full paper [27]. As in the analysis of VAMP, the LSL considers a sequence of problems indexed by the vector dimension $N$. For each $N$, we assume that there is a "true" vector $\mathbf{x}^0 \in \mathbb{R}^N$ that is observed through measurements of the form

$$\mathbf{y} = \mathbf{A}\mathbf{x}^0 + \mathbf{w} \in \mathbb{R}^N, \quad \mathbf{w} \sim \mathcal{N}(\mathbf{0}, \theta_2^{-1}\mathbf{I}_N), \tag{13}$$

where $\mathbf{A} \in \mathbb{R}^{N \times N}$ is a known transform, $\mathbf{w}$ is Gaussian noise and $\theta_2$ represents a "true" noise precision. The noise precision $\theta_2$ does not change with $N$.

Identical to [9], the transform $\mathbf{A}$ is modeled as a large, *right-orthogonally invariant* random matrix. Specifically, we assume that it has an SVD of the form (9) where $\mathbf{U}$ and $\mathbf{V}$ are $N \times N$ orthogonal matrices such that $\mathbf{U}$ is deterministic and $\mathbf{V}$ is Haar distributed (i.e. uniformly distributed on the set of orthogonal matrices). As described in [9], although we have assumed a square matrix $\mathbf{A}$, we can consider general rectangular $\mathbf{A}$ by adding zero singular values.

Using the definitions in full paper [27], we assume that the components of the singular-value vector $\mathbf{s} \in \mathbb{R}^N$ in (9) converge empirically with second-order moments as

$$\lim_{N \to \infty} \{s_n\} \overset{PL(2)}{=} S, \tag{14}$$

for some non-negative random variable $S$ with $\mathbb{E}[S] > 0$ and $S \in [0, S_{\max}]$ for some finite maximum value $S_{\max}$. Additionally, we assume that the components of the true vector, $\mathbf{x}^0$, and the initial input to the denoiser, $\mathbf{r}_{10}$, converge empirically as

$$\lim_{N \to \infty} \{(r_{10,n}, x_n^0)\} \overset{PL(2)}{=} (R_{10}, X^0), \quad R_{10} = X^0 + P_0, \quad P_0 \sim \mathcal{N}(0, \tau_{10}), \tag{15}$$

where $X^0$ is a random variable representing the *true distribution* of the components $\mathbf{x}^0$; $P_0$ is an initial error and $\tau_{10}$ is an initial error variance. The variable $X^0$ may be distributed as $X^0 \sim p(\cdot|\boldsymbol{\theta}_1)$ for some true parameter $\boldsymbol{\theta}_1$. However, in order to incorporate under-modeling, the existence of such a true parameter is not required. We also assume that the initial second-order term and parameter estimate converge almost surely as

$$\lim_{N \to \infty} (\gamma_{10}, \widehat{\boldsymbol{\theta}}_{10}, \widehat{\theta}_{20}) = (\overline{\gamma}_{10}, \overline{\theta}_{10}, \overline{\theta}_{20}) \tag{16}$$

for some $\overline{\gamma}_{10} > 0$ and $(\overline{\theta}_{10}, \overline{\theta}_{20})$.

## 3.2 Error and Sensitivity Functions

We next need to introduce parametric forms of two key terms from [9]: error functions and sensitivity functions. The error functions describe MSE of the denoiser and output estimators under AWGN measurements. Specifically, for the denoiser $g_1(\cdot, \gamma_1, \widehat{\boldsymbol{\theta}}_1)$, we define the error function as

$$\mathcal{E}_1(\gamma_1, \tau_1, \widehat{\boldsymbol{\theta}}_1) := \mathbb{E}\left[(g_1(R_1, \gamma_1, \widehat{\boldsymbol{\theta}}_1) - X^0)^2\right], \quad R_1 = X^0 + P, \quad P \sim \mathcal{N}(0, \tau_1), \tag{17}$$

where $X^0$ is distributed according to the true distribution of the components $\mathbf{x}^0$ (see above). The function $\mathcal{E}_1(\gamma_1, \tau_1, \widehat{\boldsymbol{\theta}}_1)$ thus represents the MSE of the estimate $\widehat{X} = g_1(R_1, \gamma_1, \widehat{\boldsymbol{\theta}}_1)$ from a measurement $R_1$ corrupted by Gaussian noise of variance $\tau_1$ under the parameter estimate $\widehat{\boldsymbol{\theta}}_1$. For the output estimator, we define the error function as

$$\mathcal{E}_2(\gamma_2, \tau_2, \widehat{\theta}_2) := \lim_{N \to \infty} \frac{1}{N} \mathbb{E} \|\mathbf{g}_2(\mathbf{r}_2, \gamma_2, \widehat{\theta}_2) - \mathbf{x}^0\|^2,$$

$$\mathbf{x}^0 = \mathbf{r}_2 + \mathbf{q}, \quad \mathbf{q} \sim \mathcal{N}(0, \tau_2 \mathbf{I}), \quad \mathbf{y} = \mathbf{A}\mathbf{x}^0 + \mathbf{w}, \quad \mathbf{w} \sim \mathcal{N}(0, \theta_2^{-1}\mathbf{I}), \tag{18}$$

which is the average per component error of the vector estimate under Gaussian noise. The dependence on the true noise precision, $\theta_2$, is suppressed.

The sensitivity functions describe the expected divergence of the estimator. For the denoiser, the sensitivity function is defined as

$$A_1(\gamma_1, \tau_1, \widehat{\boldsymbol{\theta}}_1) := \mathbb{E}\left[g_1'(R_1, \gamma_1, \widehat{\boldsymbol{\theta}}_1)\right], \quad R_1 = X^0 + P, \quad P \sim \mathcal{N}(0, \tau_1), \tag{19}$$

which is the average derivative under a Gaussian noise input. For the output estimator, the sensitivity is defined as

$$A_2(\gamma_2, \tau_2, \widehat{\theta}_2) := \lim_{N \to \infty} \frac{1}{N} \text{tr} \left[\frac{\partial \mathbf{g}_2(\mathbf{r}_2, \gamma_2, \widehat{\theta}_2)}{\partial \mathbf{r}_2}\right], \tag{20}$$

where $\mathbf{r}_2$ is distributed as in (18). The paper [9] discusses the error and sensitivity functions in detail and shows how these functions can be easily evaluated.

### 3.3 State Evolution Equations

We can now describe our main result, which are the SE equations for Adaptive VAMP. The equations are an extension of those in the VAMP paper [9], with modifications for the parameter estimation. For a given iteration $k \geq 1$, consider the set of components,

$$\{(\widehat{x}_{1k,n}, r_{1k,n}, x_n^0), \; n = 1, \ldots, N\}.$$

This set represents the components of the true vector $\mathbf{x}^0$, its corresponding estimate $\widehat{\mathbf{x}}_{1k}$ and the denoiser input $\mathbf{r}_{1k}$. We will show that, under certain assumptions, these components converge empirically as

$$\lim_{N \to \infty} \{(\widehat{x}_{1k,n}, r_{1k,n}, x_n^0)\} \overset{PL(2)}{=} (\widehat{X}_{1k}, R_{1k}, X^0), \tag{21}$$

where the random variables $(\widehat{X}_{1k}, R_{1k}, X^0)$ are given by

$$R_{1k} = X^0 + P_k, \quad P_k \sim \mathcal{N}(0, \tau_{1k}), \quad \widehat{X}_{1k} = g_1(R_{1k}, \overline{\gamma}_{1k}, \overline{\theta}_{1k}), \tag{22}$$

for constants $\overline{\gamma}_{1k}$, $\overline{\theta}_{1k}$ and $\tau_{1k}$ that will be defined below. We will also see that $\widehat{\boldsymbol{\theta}}_{1k} \to \overline{\theta}_{1k}$, so $\overline{\theta}_{1k}$ represents the asymptotic parameter estimate. The model (22) shows that each component $r_{1k,n}$ appears as the true component $x_n^0$ plus Gaussian noise. The corresponding estimate $\widehat{x}_{1k,n}$ then appears as the denoiser output with $r_{1k,n}$ as the input and $\overline{\theta}_{1k}$ as the parameter estimate. Hence, the asymptotic behavior of any component $x_n^0$ and its corresponding $\widehat{x}_{1k,n}$ is identical to a simple scalar system. We will refer to (21)-(22) as the denoiser's *scalar equivalent model*.

We will also show that these transformed errors $\mathbf{q}_k$ and noise $\boldsymbol{\xi}$ in (10) and singular values $\mathbf{s}$ converge empirically to a set of independent random variables $(Q_k, \Xi, S)$ given by

$$\lim_{N \to \infty} \{(q_{k,n}, \xi_n, s_n)\} \overset{PL(2)}{=} (Q_k, \Xi, S), \quad Q_k \sim \mathcal{N}(0, \tau_{2k}), \quad \Xi \sim \mathcal{N}(0, \theta_2^{-1}), \tag{23}$$

where $S$ has the distribution of the singular values of $\mathbf{A}$, $\tau_{2k}$ is a variance that will be defined below and $\theta_2$ is the true noise precision in the measurement model (13). All the variables in (23) are independent. Thus (23) is a scalar equivalent model for the output estimator.

The variance terms are defined recursively through the *state evolution* equations,

$$\overline{\alpha}_{1k} = A_1(\overline{\gamma}_{1k}, \tau_{1k}, \overline{\theta}_{1k}), \quad \overline{\eta}_{1k} = \frac{\overline{\gamma}_{1k}}{\overline{\alpha}_{1k}}, \quad \overline{\gamma}_{2k} = \overline{\eta}_{1k} - \overline{\gamma}_{1k} \tag{24a}$$

$$\overline{\theta}_{1,k+1} = T_1(\overline{\mu}_{1k}), \quad \overline{\mu}_{1k} = \mathbb{E}\left[\phi_1(R_{1k}, \overline{\gamma}_{1k}, \overline{\theta}_{1k})\right] \tag{24b}$$

$$\tau_{2k} = \frac{1}{(1 - \overline{\alpha}_{1k})^2} \left[\mathcal{E}_1(\overline{\gamma}_{1k}, \tau_{1k}, \overline{\theta}_{1k}) - \overline{\alpha}_{1k}^2 \tau_{1k}\right], \tag{24c}$$

$$\overline{\alpha}_{2k} = A_2(\overline{\gamma}_{2k}, \tau_{2k}, \overline{\theta}_{2k}), \quad \overline{\eta}_{2k} = \frac{\overline{\gamma}_{2k}}{\overline{\alpha}_{2k}}, \quad \overline{\gamma}_{1,k+1} = \overline{\eta}_{2k} - \overline{\gamma}_{2k} \tag{24d}$$

$$\overline{\theta}_{2,k+1} = T_2(\overline{\mu}_{2k}), \quad \overline{\mu}_{2k} = \mathbb{E}\left[\phi_2(Q_k, \Xi, S, \overline{\gamma}_{2k}, \overline{\theta}_{2k})\right] \tag{24e}$$

$$\tau_{1,k+1} = \frac{1}{(1 - \overline{\alpha}_{2k})^2} \left[\mathcal{E}_2(\overline{\gamma}_{2k}, \tau_{2k}) - \overline{\alpha}_{2k}^2 \tau_{2k}\right], \tag{24f}$$

which are initialized with $\tau_{10} = \mathbb{E}[(R_{10} - X^0)^2]$ and the $(\overline{\gamma}_{10}, \overline{\theta}_{10}, \overline{\theta}_{20})$ defined from the limit (16). The expectation in (24b) is with respect to the random variables (21) and the expectation in (24e) is with respect to the random variables (23).

**Theorem 1.** *Consider the outputs of Algorithm 1. Under the above assumptions and definitions, assume additionally that for all iterations $k$:*

(i) *The solution $\overline{\alpha}_{1k}$ from the SE equations (24) satisfies $\overline{\alpha}_{1k} \in (0, 1)$.*

(ii) *The functions $A_i(\cdot)$, $\mathcal{E}_i(\cdot)$ and $T_i(\cdot)$ are continuous at $(\gamma_i, \tau_i, \widehat{\theta}_i, \mu_i) = (\overline{\gamma}_{ik}, \tau_{ik}, \overline{\theta}_{ik}, \overline{\mu}_{ik})$.*

(iii) *The denoiser function $g_1(r_1, \gamma_1, \widehat{\boldsymbol{\theta}}_1)$ and its derivative $g_1'(r_1, \gamma_1, \widehat{\boldsymbol{\theta}}_1)$ are uniformly Lipschitz in $r_1$ at $(\gamma_1, \widehat{\boldsymbol{\theta}}_1) = (\overline{\gamma}_{1k}, \overline{\theta}_{1k})$. (See the full paper [27]. for a precise definition of uniform Lipschitz continuity.)*

*(iv) The adaptation statistic $\phi_1(r_1, \gamma_1, \widehat{\boldsymbol{\theta}}_1)$ is uniformly pseudo-Lipschitz of order 2 in $r_1$ at $(\gamma_1, \widehat{\boldsymbol{\theta}}_1) = (\overline{\gamma}_{1k}, \overline{\theta}_{1k})$.*

*Then, for any fixed iteration $k \geq 0$,*

$$\lim_{N \to \infty} (\alpha_{ik}, \eta_{ik}, \gamma_{ik}, \mu_{ik}, \widehat{\theta}_{ik}) = (\overline{\alpha}_{ik}, \overline{\eta}_{ik}, \overline{\gamma}_{ik}, \overline{\mu}_{ik}, \overline{\theta}_{ik}) \tag{25}$$

*almost surely. In addition, the empirical limit (21) holds almost surely for all $k > 0$, and (23) holds almost surely for all $k \geq 0$.*

Theorem 1 shows that, in the LSL, the parameter estimates $\widehat{\theta}_{ik}$ converge to deterministic limits $\overline{\theta}_{ik}$ that can be precisely predicted by the state-evolution equations. The SE equations incorporate the true distribution of the components on the prior $\mathbf{x}^0$, the true noise precision $\theta_2$, and the specific parameter estimation and denoiser functions used by the Adaptive VAMP method. In addition, similar to the SE analysis of VAMP in [9], the SE equations also predict the asymptotic joint distribution of $\mathbf{x}^0$ and their estimates $\widehat{\mathbf{x}}_{ik}$. This joint distribution can be used to measure various performance metrics such as MSE – see [9]. In this way, we have provided a rigorous and precise characterization of a class of adaptive VAMP algorithms that includes EM-VAMP.

## 4 Consistent Parameter Estimation with Variance Auto-Tuning

By comparing the deterministic limits $\overline{\theta}_{ik}$ with the true parameters $\theta_i$, one can determine under which problem conditions the parameter estimates of adaptive VAMP are asymptotically consistent. In this section, we show with a particular choice of parameter estimation functions, one can obtain provably asymptotically consistent parameter estimates under suitable identifiability conditions. We call the method *variance auto-tuning*, which generalizes the approach in [7].

**Definition 1.** *Let $p(\mathbf{x}|\boldsymbol{\theta}_1)$ be a parametrized set of densities. Given a finite-dimensional statistic $\phi_1(r)$, consider the mapping*

$$(\tau_1, \boldsymbol{\theta}_1) \mapsto \mathbb{E}\left[\phi_1(R)|\tau_1, \boldsymbol{\theta}_1\right], \quad R = X + \mathcal{N}(0, \tau_1), \quad X \sim p(x|\boldsymbol{\theta}_1). \tag{26}$$

*We say the $p(\mathbf{x}|\boldsymbol{\theta}_1)$ is* identifiable *in Gaussian noise if there exists a finite-dimensional statistic $\phi_1(r) \in \mathbb{R}^d$ such that (i) $\phi_1(r)$ is pseudo-Lipschitz continuous of order 2; and (ii) the mapping (26) has a continuous inverse.*

**Theorem 2.** *Under the assumptions of Theorem 1, suppose that $X^0$ follows $X^0 \sim p(\mathbf{x}|\boldsymbol{\theta}_1^0)$ for some true parameter $\boldsymbol{\theta}_1^0$. If $p(\mathbf{x}|\boldsymbol{\theta}_1)$ is identifiable in Gaussian noise, there exists an adaptation rule such that, for any iteration $k$, the estimate $\widehat{\boldsymbol{\theta}}_{1k}$ and noise estimate $\widehat{\tau}_{1k}$ are asymptotically consistent in that $\lim_{N \to \infty} \widehat{\boldsymbol{\theta}}_{1k} = \boldsymbol{\theta}_1^0$ and $\lim_{N \to \infty} \widehat{\tau}_{1k} = \tau_{1k}$ almost surely.*

The theorem is proved in full paper [27]. which also provides details on how to perform the adaptation. A similar result for consistent estimation of the noise precision $\theta_2$ is also given. The result is remarkable as it shows that a simple variant of EM-VAMP can provide provably consistent parameter estimates under extremely general distributions.

## 5 Numerical Simulations

**Sparse signal recovery:** The paper [8] presented several numerical experiments to assess the performance of EM-VAMP relative to other methods. Here, our goal is to confirm that EM-VAMP's performance matches the SE predictions. As in [8], we consider a sparse linear regression problem of estimating a vector $\mathbf{x}$ from measurements $\mathbf{y}$ from (1) without knowing the signal parameters $\boldsymbol{\theta}_1$ or the noise precision $\theta_2 > 0$. Details are given in the full paper [27]. Briefly, to model the sparsity, $\mathbf{x}$ is drawn as an i.i.d. Bernoulli-Gaussian (i.e., spike and slab) prior with unknown sparsity level, mean and variance. The true sparsity is $\beta_x = 0.1$. Following [15, 16], we take $\mathbf{A} \in \mathbb{R}^{M \times N}$ to be a random right-orthogonally invariant matrix with dimensions under $M = 512$, $N = 1024$ with the condition number set to $\kappa = 100$ (high condition number matrices are known to be problem for conventional AMP methods). The left panel of Fig. 1 shows the normalized mean square error (NMSE) for various algorithms. The full paper [27] describes the algorithms in details and also shows similar results for $\kappa = 10$.

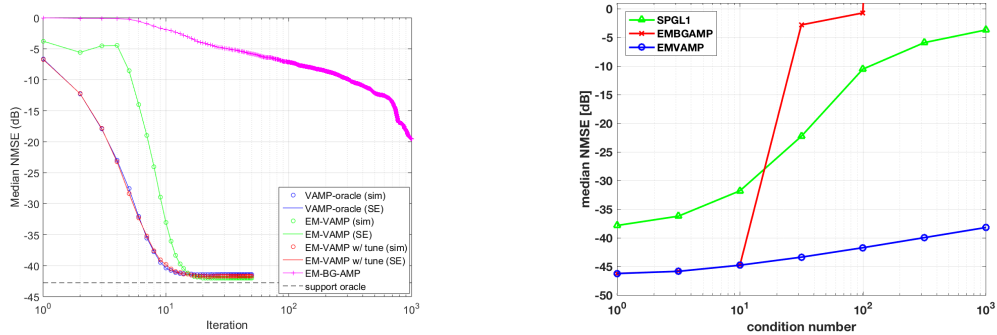

Figure 1: Numerical simulations. Left panel: Sparse signal recovery: NMSE versus iteration for condition number for a random matrix with a condition number $\kappa = 100$. Right panel: NMSE for sparse image recovery as a function of the measurement ratio $M/N$.

We see several important features. First, for all variants of VAMP and EM-VAMP, the SE equations provide an excellent prediction of the per iteration performance of the algorithm. Second, consistent with the simulations in [9], the oracle VAMP converges remarkably fast ($\sim 10$ iterations). Third, the performance of EM-VAMP with auto-tuning is virtually indistinguishable from oracle VAMP, suggesting that the parameter estimates are near perfect from the very first iteration. Fourth, the EM-VAMP method performs initially worse than the oracle-VAMP, but these errors are exactly predicted by the SE. Finally, all the VAMP and EM-VAMP algorithm exhibit much faster convergence than the EM-BG-AMP. In fact, consistent with observations in [8], EM-BG-AMP begins to diverge at higher condition numbers. In contrast, the VAMP algorithms are stable.

**Compressed sensing image recovery**    While the theory is developed on theoretical signal priors, we demonstrate that the proposed EM-VAMP algorithm can be effective on natural images. Specifically, we repeat the experiments in [28] for recovery of a sparse image. Again, see the full paper [27] for details including a picture of the image and the various reconstructions. An $N = 256 \times 256$ image of a satellite with $K = 6678$ pixels is transformed through an undersampled random transform $\mathbf{A} = \text{diag}(\mathbf{s})\mathbf{PH}$, where $\mathbf{H}$ is fast Hadamard transform, $\mathbf{P}$ is a random subselection to $M$ measurements and $\mathbf{s}$ is a scaling to adjust the condition number. As in the previous example, the image vector $\mathbf{x}$ is modeled as a sparse Bernoulli-Gaussian and the EM-VAMP algorithm is used to estimate the sparsity ratio, signal variance and noise variance. The transform is set to have a condition number of $\kappa = 100$. We see from the right panel of Fig. 1 we see that the that the EM-VAMP algorithm is able to reconstruct the images with improved performance over the standard basis pursuit denoising method spgl1 [29] and the EM-BG-GAMP method from [16].

## 6    Conclusions

Due to its analytic tractability, computational simplicity, and potential for Bayes optimal inference, VAMP is a promising technique for statistical linear inverse problems. However, a key challenge in using VAMP and related methods is the need to precisely specify the distribution on the problem parameters. This work provides a rigorous foundation for analyzing VAMP in combination with various parameter adaptation techniques including EM. The analysis reveals that VAMP with appropriate tuning, can also provide consistent parameter estimates under very general settings, thus yielding a powerful approach for statistical linear inverse problems.

**Acknowledgments**

A. K. Fletcher and M. Saharee-Ardakan were supported in part by the National Science Foundation under Grants 1254204 and 1738286 and the Office of Naval Research under Grant N00014-15-1-2677. S. Rangan was supported in part by the National Science Foundation under Grants 1116589, 1302336, and 1547332, and the industrial affiliates of NYU WIRELESS. The work of P. Schniter was supported in part by the National Science Foundation under Grant CCF-1527162.

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
