[Reviews · NeurIPS 2017]

Reviewer 1



This paper proposes a novel method to estimate a random vector x from noisy linear measurements as well as estimating the unknown parameters on the distributions of x and w. The authors provide provable guarantees on the performance of the algorithm. The proposed approach is based on approximate message passing techniques and extends previous results in this area. I was able to follow the paper and find the results very interesting, especially the ability to materialistically predict the performance based on state evolution equations. I know that the proposed approach is of interest for many learning techniques, but since I am not an expert in this field I cannot be a good judge on how the proposed algorithm stands w.r.t. other similar methods in this area.

Reviewer 2



This paper extends the vector approximate message passing framework to a linear problem involving both inference (on the cause vector x) and parameter estimation (on the prior over x and the noise process over the observed y). VAMP was developed in a similar setting, but without learning over parameters. As such, some of the technical content is similar to the original paper (the inference part of VAMP, the state evolution characterization from theorem 1, the proof technique), but there is novel content related to the parameter estimation (the adaptive part of VAMP, and the consistency proof of theorem 2). The presentation is not always very clear (the paper could use a picture of the graphical model, and some table of notation with corresponding meaning for the different variables). Numerical experiments demonstrate strong performance of the algorithm compared to standard algorithms for sparse signal recovery.

Reviewer 3



The model is p(y | x, \theta_1) p(x | \theta_1) where y | x is a linear system y = A*x + Gaussian noise. We want to infer x and \theta from data. Importantly, p(x | \theta_1) factorizes over elements of x. The paper takes EM-VAMP from [13], and changes a step for updating \theta_1 (from the EM-VAMP algorithm). As a general comment, I was wondering why the algorithm wasn't called "generalized EM-VAMP" -- if eq (7) is used as "adaptation" (which is preferred according to line 113), then there is no difference and the naming is rather confusing. What is adapted that is *not* adapted in EM-VAMP? Naming and nomenclature aside, and also ignoring that much of the introduction of the paper is a verbatim copy of [13], the paper makes some fundamental contributions: It provides an asymptotic analysis to EM-VAMP, showing that as the number of latent dimensions in x goes to infinity, the algorithm converges. Secondly, we can compute the deterministic limit from a set of state evolution equations. The paper's numerical results show that the state evolution equations in eq (24) in the paper predicts the performance (mean squared error of true x vs estimated x^) of EM-VAMP remarkably well per iteration. The paper then asks, given that the algorithm converges, does it converge to an asymptotically consistent estimate of \theta? If p(x|\theta_1) is identifiable in Gaussian noise, then we can find statistics (essentially, here the "adaptation rule" that generalizes EM-VAMP seems to really come into play) so that convergence of the estimate is a.s. to \theta. In my humble opinion, the paper provides nice and rigorous analysis to give EM-VAMP a deeper theoretic base. Postscripts ----------- Very similar to [a] and [b], the algorithm factorizes the joint into a Gaussian and a non-Gaussian part, and approximates the non-Gaussian part with a contribution (messages) towards the Gaussian part to give a Gaussian belief on x. [a] Matthias Seeger & Hannes Nickisch, Fast Convergent Algorithms for Expectation Propagation Approximate Bayesian Inference. [b] Matthias Seeger, Bayesian Inference and Optimal Design for the Sparse Linear Model, JMLR Could the authors put Figure 1 in a vectorized format? It is hard to read, especially when printed. small typo, :== in line 508

Reviewer 4



This paper studies the classical problem of estimating a random vector x from linear measurements y = Ax + w where A is known and w is Gaussian noise. When the matrix A is well-behaved (i.e. zero-mean sub-Gaussian iid), a well-established method for this problem is approximate message passing (AMP), a fast iterative algorithm. Prior work has extended this to the VAMP (Vector AMP) algorithm, which works for a much larger class of matrices A (right-rotationally invariant) which includes arbitrarily ill-conditioned matrices. Rigorous results for AMP and VAMP are already known; in particular, their performance is described by simple scalar recurrences known as "state evolution." This paper extends the rigorous analysis of VAMP to include the case of unknown parameters. In particular, each component of x is drawn from some distribution which depends on parameters theta_1, and the noise w is iid Gaussian with variance 1/theta_2. Prior work has introduced the EM-VAMP algorithm which combines VAMP with EM in order to estimate the unknown parameters theta_1, theta_2; but until now, no rigorous guarantees have been given for this algorithm. The current paper introduces the "Adaptive VAMP" algorithm (generalizing EM-VAMP) and proves a rigorous state evolution recurrence that describes both estimates of x and estimates of the unknown parameters. Using this, they prove that a particular case of the algorithm gives consistent estimates of the parameters under certain conditions. They also show that the algorithm gives Bayes-optimal estimates for x in some settings. Numerical simulations support the theoretical results of the paper. They also show that the algorithm works well on real image data. Many parts of the proof are not novel (following from a minor modification of prior work). However, one novel idea is "variance auto-tuning," the update step that allows for provably consistent estimation of the parameters. The idea is roughly that the algorithm estimates how much uncertainty there is in its current estimate for x, and uses this value in the next iteration. Overall I think this paper is a solid contribution to the theory of AMP, and I recommend its acceptance.